# TET3 is a positive regulator of mitochondrial respiration in Neuro2A cells

**Valeria Leon Kropf, Caraugh J. Albany, Anna Zoccarato◉, Hannah L. H. Green◉, Youwen Yang, Alison C. Brewer◉ ***

School of Cardiovascular and Metabolic Medicine & Sciences, King's College London British Heart Foundation Centre of Excellence, London, United Kingdom

* alison.brewer@kcl.ac.uk

**Data Availability Statement:** All relevant data files are accessible via the SRA at NCBI under BioProject PRJNA1035523 (link: https://www.ncbi.nlm.nih.gov/sra/PRJNA1035523).

## Abstract

Ten-Eleven-Translocase (TET) enzymes contribute to the regulation of the methylome *via* successive oxidation of 5-methyl cytosine (5mC) to derivatives which can be actively removed by base-excision-repair (BER) mechanisms in the absence of cell division. This is particularly important in post-mitotic neurons where changes in DNA methylation are known to associate with changes in neural function. TET3, specifically, is a critical regulator of both neuronal differentiation in development and mediates dynamic changes in the methylome of adult neurons associated with cognitive function. While DNA methylation is understood to regulate transcription, little is known of the specific targets of TET3-dependent catalytic activity in neurons. We report the results of an unbiased transcriptome analysis of the neuroblastoma-derived cell line; Neuro2A, in which Tet3 was silenced. Oxidative phosphorylation (OxPhos) was identified as the most significantly down-regulated functional canonical pathway, and these findings were confirmed by measurements of oxygen consumption rate in the Seahorse bioenergetics analyser. The mRNA levels of both nuclear- and mitochondrial-encoded OxPhos genes were reduced by Tet3-silencing, but we found no evidence for differential (hydroxy)methylation deposition at these gene loci. However, the mRNA expression of genes known to be involved in mitochondrial quality control were also shown to be significantly downregulated in the absence of TET3. One of these genes; EndoG, was identified as a direct target of TET3-catalytic activity at non-CpG methylated sites within its gene body. Accordingly, we propose that aberrant mitochondrial homeostasis may contribute to the decrease in OxPhos, observed upon Tet3-downregulation in Neuro2A cells.

## Introduction

DNA methylation is a stable epigenetic covalent modification which is now recognised to be an important regulator of gene transcription [1]. In mammals, DNA methylation occurs at the C5 position of cytosine (5methylcytosine[5mC]), predominantly within the 5'CpG3' dinucleotide sequence [2]. DNA methylation patterns can be established *de novo* and maintained at cell division by the enzymatic actions of DNA methyltransferases (DNMTs). Loss of methylation can thus occur "passively" if this maintenance methylation is lacking during rounds of DNA

**Funding:** This work was supported by the National Program of Scholarships and Student Loans (PRONABEC), Minister of Education, Peru (Law No 29837, Supreme Decree 013-2012-ED, Supreme Decree 008-2013-ED, Supreme Decree 01–2015-MINEDU, RJ No 4320-2018-MINEDU-VMGI-PRONABEC to VLK. The sponsors played no role in the study design, data collection and analysis, decision to publish, or preparation of the manuscript.

**Competing interests:** The authors have declared that no competing interests exist.

replication [3]. Alternatively, active demethylation mechanisms exist which involve a family of three DNA hydroxylases, termed Ten Eleven Translocase enzymes (TET1-3). These are members of the 2-oxoglutarate dependent dioxygenase (2-OGDD) superfamily which act to convert 5mC into 5-hydroxymethylcytosine (5hmC), 5-formylcytosine (5fC) and 5-carboxylmethylcytosine (5caC) *via* successive oxidation reactions [4]. 5fC and 5caC can subsequently be excised and replaced with unmodified cytosine *via* the action of thymine DNA glycosylase (TDG) coupled to base excision repair (BER) mechanisms. 5hmC is the most stable of these modifications and consequently can accumulate to significant levels, where it is recognised to play specific regulatory functions, most notably in the brain [5].

The expression and functional significance of TET family members and associated dynamics of 5hmC deposition have been investigated in both differentiating neuronal tissue and in adult, post mitotic neurons. 5hmC levels were shown to increase during embryonic mouse brain development and to become enriched within gene bodies of activated neuronal function-related genes [6]. TET3 expression, specifically, has been shown to be induced in embryonic stem cells (ESCs) *in vitro*, during neuronal differentiation and to be functionally necessary for the maintenance and terminal differentiation of neuronal progenitor cells [7]. In the adult mammalian brain, an increasing number of studies have identified TET3, associated with dynamic changes in 5hmC, as playing critical roles in neural function. TET3 is the most highly expressed of the TET enzymes in specific regions of the brain which display correspondingly high levels of 5hmC; the cerebellum, cortex and hippocampus [8]. Further, functional behavioural studies in mice have identified TET3 to be important in behavioural adaptation processes, such as fear extinction, anxiety, learning and memory, which are often known to be associated with changes in 5hmC accumulation in post-mitotic neurons [9–12].

Consistent with the clear importance of the dynamic regulation of 5hmC in the normal development and physiology of the brain, aberrant global and/or locus-specific 5hmC patterns associate with a broad range of neurodevelopmental abnormalities (including autism and Rett syndrome [13]) and age-related neurodegenerative disorders (including Alzheimer's disease, Huntington's disease, Parkinson's disease and schizophrenia [14–20]). The enzymes which mediate this epigenetic modification, including TET3, thus present attractive potential therapeutic targets to mitigate neural disorders. In addition, in common with other epigenetic modifiers, including histone modifiers and other TET proteins, TET3 is known to regulate transcription *via* non-enzymatic mechanisms [21, 22]. An understanding of the molecular mechanisms which regulate the activity and specificity of TET3-mediated DNA hydroxylation and/or non-catalytic mechanisms in neuronal cells and the downstream biochemical functions impacted by TET3 activity is therefore essential. To begin to address this, we therefore investigated the transcriptional and functional roles of TET3 in the neuroblastoma cell line, Neuro2A (N2A), and have here identified its importance in the regulation of mitochondrial function and respiration. We also identify the mitochondrion-localised endonuclease, EndoG [23], as a putative direct target of Tet3 catalytic activity and transcriptional regulation in neuronal cells. Based on current literature, we suggest that EndoG may impact upon mitochondrial function in neuronal cells and may therefore in part mediate the TET3-dependent regulation of mitochondrial respiration.

## Materials and methods

### N2A cell culture

N2A cells were cultured in (low glucose; 1000g/l; approximately 5.6mM) DMEM supplemented with FBS (10%) and PenStrep/L-glutamine and split (1:10) every 3 days. Ascorbic acid (50μM), dimethyloxalylglycine (DMOG; 1mM) and 5-azacytidine (5-azaC, 1μM; replaced

every 24 hours) and appropriate vehicle controls were added 24 hours after plating and cells harvested after 24, 48 or 72 hours respectively. siRNA transfection was carried out 24 hours after plating with SMARTpool: ON-TARGETplus TET3 siRNA (Horizon) or Silencer™ Select Negative Control No. 1 siRNA (Thermo Fisher) using RNAiMAX Transfection Reagent (Thermo Fisher) according to manufacturer's instructions.

## RNA-seq and QPCR analyses

RNA was extracted using the ReliaPrep RNA Tissue Miniprep System (Promega) according to the manufacturer's instructions. Triplicate samples of RNA were pooled in equal amounts to a final quantity of 1µg of RNA in 25µl nuclease-free H2O. Library preparation, sequencing (20 million sequence reads) and initial bioinformatic analysis was conducted by BGI Genomics Co. (China) using the DNBseq™ sequencing platform. Further analysis was performed using Ingenuity Pathway Analysis (IPA) software (Qiagen) applying the threshold of false discovery rate (FDR) <0.05.

cDNA was synthesised from 500 ng of RNA using LunaScript® RT Master Mix Kit (New England Biolabs) according to the manufacturer's instructions. Relative gene expression was quantified using SYBR green fluorescent dye (PCR Biosystems). Samples were amplified using a StepOnePlus™ Real-Time PCR system (Applied Biosystems). mRNA expression was analysed using the $\Delta$ Ct method. A list of primer sequences used is given in S1 Table in S1 File.

## Western blotting

Cells were scraped into NP-40 lysis buffer (25 mM Tris-HCl (pH 7.4), 150 mM NaCl, 2 mM EGTA, 5 mM EDTA, 0.5% (v/v) NP-40, 30 mM sodium fluoride, 40 mM β- glycerophosphate, 20 mM sodium pyrophosphate, 1 mM sodium orthovanadate, 1 mM phyenylmethylsulfonyl-fluoride), supplemented with protease inhibitor cocktail II (Sigma, #P5726), cocktail III (Sigma, #P0044) and phosphatase inhibitor (Sigma, #P3840). Lysates were sonicated X2 for 5 sec (Branson 150 sonifier, setting 1). Protein concentration was determined using Pierce™ BCA Protein Assay Kit (Thermo Fisher) according to the manufacturer's instructions. Protein aliquots (20µg) were electrophoresed on a precast 4 to 12% gradient Bolt™ Bis-Tris, 1.0 mm polyacrylamide gel (Thermo Fisher) and transferred onto nitrocellulose membranes (GE Healthcare).

Membranes were blocked in 5% milk TBS/T and probed overnight at 4°C with the OxPhos antibody cocktail (Abcam; dilution as directed by supplier), anti-ENDOG (Abcam: ab76122; 1:1000), anti-α-Tubulin (Sigma: T5168, 1:5000) or anti-β-Actin (Sigma; 1:5000). Secondary Antibodies (LiCOR; 1:15,000) were subsequently incubated for 1hr at room temperature. Blots were visualised using an Odyssey CLX scanner (LiCOR) and quantified using Image Studio™ software (LiCor).

## Oxygen consumption rate (OCR) quantification

OCR in live N2A cells was measured on an extracellular flux analyzer (XFe24 Seahorse; Agilent Technologies), essentially according to the manufacturer's protocol for mitochondrial stress tests, with the modification of 3 measurement cycles for basal OCR and 2 for other conditions, each composed of 1.5 minutes of mixing, 2 minutes of waiting and 2 minutes of measuring. The concentrations of drugs, added as indicated in Fig 2A, were oligomycin (1µM, Sigma, 75351), oligo cyanide-4-(trifluoromethoxy) phenylhydrazone (FCCP) (2 µM Sigma, C2920), rotenone (1 µM, Sigma, A8674), antimycin A (1 µM, Sigma, R8875). At the end of all measurements, cells were fixed with 2% paraformaldehyde (10 mins at room temperature) and incubated with DRAQ5™ (Thermo Fisher) (1:1000) for 20 min at room temperature. Cell numbers

were measured using an Odyssey CLX scanner and quantified using Image Studio™ software. OCR measurements were normalised to cell number.

## DNA extraction and hMeDIP-seq

N2a cells were harvested and re-suspended in digestion buffer (150 mM NaCl, 1% SDS, 5 mM EDTA, 50 mM Tris-HCl (pH 8)) with Proteinase K (350 µg/ml) overnight at 55˚C. Samples were then incubated with ribonuclease A (140 µg/ml) at 55˚C for 30 min followed by standard phenol/chloroform extraction and isopropanol precipitation. Genomic DNA from duplicate Tet3-silenced and control N2A cells were submitted to Arraystar Inc. (Rockville, Maryland) where hMeDIP enrichment and NGS processing and bioinformatic analyses were performed as described [24]. The sequencing quality (Q30) was >78% in all cases, and >77% of all reads were mapped in all samples. hMeDIP-enriched regions (peaks) were annotated by the nearest gene using the UCSC RefSeq database (https://genome.ucsc.edu/). Differentially hydroxy-methylated regions (DhMRs) displaying statistically significant differences within specific genetic elements (promoters, enhancers and non-coding RNAs) between two groups were identified by diffReps (Cut-off: log2FC≥1, p-value≤10−3) to control for false positives. The enhancer-and super-enhancer associated gene information (used for Gene Ontology analyses) was collected from VISTA/FANTOM.

## Single base resolution analysis of 5hmC

5hmC deposition at the base resolution was performed by its selective enzyme-based conversion to glucosylated-hydroxymethylcytosine and subsequent protection from deamination by APOBEC [25]. Triplicate genomic N2A DNA samples were pooled and sonicated using a Bandelin Sonorex water bath sonicator for 2x 8 min to shear the DNA. 1 µg of each sample was mixed with 2U of T4 Phage β-glucosyltransferase (T4-BGT) (New England Biolabs) according to the manufacturer's instructions. 4 µl of formamide was then added to 200 ng of the samples and incubated at 85˚C for 10 min. Subsequently, APOBEC (New England Biolabs) was added according to the manufacturer's instructions to deaminate cytosines to uracils. DNA was then purified using Purification Beads (New England Biolabs) according to the manufacturer's instructions.

20 ng of each purified sample was then amplified using GoTaq® (Promega) and the gene-specific converted primers designed to amplify EpiP1 and EpiP2, as depicted in Fig 4. Primer sequences (given 5'-3') EpiP1: forward `TGGGGTTTTGTTTTGTTTAAAAGTA`, and reverse `CTCAACTACTCCAACACCCAAAA` EpiP2: forward `GGGAGGTTGTTATTTTTGTTTTTGA` and reverse `AAAACCTTCCTAACTTCCAACAAA` were designed using the EpiDesigner primer design tool (https://epidesigner.com/). Amplified regions were sequenced by Sanger Sequencing (Source BioScience) with PCR amplification primers.

## Mitochondrial number and integrity

Mitochondrial number was determined by QPCR analyses of genomic DNA to assess the relative levels of mtCO3 (mitochondrially encoded cytochrome C oxidase III) using the following primers (given 5'-3'): forward `CAAGGCCACCACACTCCTAT` and reverse `GTCAGCAGCCT CCTAGATC`. These were normalised to (nuclear) genomic DNA levels of mouse α-albumin intron 2 using the following primers (given 5'-3'): forward: `GGGAACCCTGGATCAACAGG` and reverse: `TCCCTACTGACTGGGGATGG`.

mtDNA integrity was determined as described previously through a QPCR-based assay in which the relative amount of a 10kb mtDNA amplicon is taken to be inversely proportional to DNA damage [26].

## Mitochondrial membrane potential (ΔΨm)

ΔΨm was measured using tetramethylrhodamine ethyl ester, perchlorate (TMRE; 200nM) fluorescence (Sigma, 87917). Five replicate cell treatments were compared to duplicate untreated N2a cells. After harvesting, cells were incubated with staining agents for 20 mins at 37°C protected from light. Cellular viability was assessed using eBioscience™ Fixable Viability Dye eFluor™ (BD Horizon™, 65-0865-14) as per the manufacture's instruction. Unstained samples and appropriate fluorescence minus one (FMO) samples were used to discriminate the negative and positive gates (S1 Fig in S1 File). FCCP (20μM) was used as a negative control (S1 Fig in S1 File). Samples were acquired on a BD LSRFortessa™ SORP FACS instrument. A minimum of 10,000 live cells were recorded. Data were analysed using the FlowJo 10.6.2 software (Tree Star Inc., USA).

## Statistical analysis

Statistical analysis were performed with Student's t-test or two-way ANOVA followed by Tukey's multiple comparison test using GraphPad Prism 9 software. All individual data points are shown and data means ± standard error of the mean (S.E.M.) are presented. Differences between data groups were considered to be statistically significant when $p < .05$.

## Results

### TET3 regulates the transcription of mitochondrial-associated proteins in N2A cells

We performed transcriptome analyses on pooled, triplicate biological samples of (undifferentiated) N2A cells to assess the transcriptional consequences of siRNA-mediated depletion of Tet3 expression in these cells. The efficient downregulation (>75%) of Tet3 was confirmed by QPCR in both the samples used in the RNA-seq analyses and in those used for the subsequent validation (Fig 1, panels A and C). The genetic depletion of Tet3 had no significant effect upon the expression of Tet2 mRNA, while the (much lower) level of endogenous Tet1 expression was only slightly, (albeit statistically significantly) upregulated and was therefore considered unlikely to compensate for the loss of TET3 activity (Fig 1A and see also discussion). In addition, a decrease in global 5hmC levels, upon TET3-depletion in N2A cells was confirmed (S2 Fig in S1 File)

The RNA-seq analyses identified 304 gene transcripts which were upregulated significantly and 172 which were downregulated significantly upon Tet3-silencing (p<0.0001; see S1 and S2 Tables in S1 File respectively) and these data are available at NCBI under GEO accession number PRJNA1035523. To identify functional pathways which might be regulated by TET3 in these neuronal cells, we analysed differentially-expressed genes, identified using a (less stringent) differential expression cut off of false discovery rate (FDR) <0.05, using the Ingenuity Pathway Analysis (IPA; Qiagen) software. This identifies canonical pathways and calculates corresponding activation z-scores for these pathways, based on both the number of genes identified within a pathway and the direction and magnitude of their change, where an absolute z score of >2 is considered significant. "Oxidative Phosphorylation" (OxPhos) was identified as the most significantly downregulated canonical pathway (z-score = -7.147, p = 5.19E$^{-12}$). The other significantly downregulated pathways (with a z-score of <-2 and p<E$^{-06}$) identified were TCA cycle (z-score = -4.123, p = 1.78E$^{-07}$), Superpathway of inositol phosphate compounds (z = -2.593, p = 1.13E$^{-07}$), Phosphoinositide biosynthesis (z = -2.286 p = 1.92 E$^{-07}$), IGF1 signalling (z = -2.4 p = 4.94E$^{-07}$) and Estrogen Receptor signalling (z = -2.069 p = 5.09 E$^{-07}$). The

only upregulated canonical pathway identified which fitted these significance criteria was EIF signalling (z = 3.543 p = $3.84E^{-23}$).

Of the 109 known OxPhos genes which are investigated in this IPA canonical pathway analysis, 54 genes were found to be downregulated (using the FDR <0.05 cut off) in our screen, while only one gene was upregulated. In addition, 13 of the genes in the OxPhos pathway are encoded in the mitochondrial genome [27] and would therefore not be identified in our transcriptome analysis (as RNA-seq reads were aligned only to the nuclear genome; see Fig 1B). The changes in the transcriptional expression of a selection of these (nuclear-expressed) OxPhos genes were confirmed to be significant by QPCR (Fig 1C). Further, the mRNA levels of a number of mitochondrial-expressed (OxPhos) genes were also assessed and found to be significantly reduced in N2A cells after Tet3-silencing (Fig 1D).

In addition to these OxPhos gene transcripts, it was evident that many other genes which were downregulated significantly (>2 fold, p<0.0001) in our RNA-seq screen are involved in the regulation of mitochondrial function, structure and/or quality control (Fig 1E). Again, the downregulation of some of these genes was validated by QPCR (Fig 1F).

## Tet3-silencing leads to reduced oxidative phosphorylation in N2A cells

The TET3-dependent changes in OxPhos gene transcription led us to investigate the effect of Tet3-silencing upon the cellular oxygen consumption in a Seahorse Bioanalyser. The basal oxygen consumption rate (OCR), spare respiratory consumption (SCR) and maximal respiration were all reduced significantly in the Tet3-silenced N2A cells (Fig 2A). The number of mitochondria/cell (as judged by the mitochondrial: nuclear DNA content) was unchanged and further there was no change evident in the integrity of the mitochondrial DNA (Fig 2B and 2C). However, the mitochondrial membrane potential (ΔΨm), as determined by the accumulation of TMRE in the mitochondrial membrane, was significantly reduced in the TET3-depleted cells (Fig 2D and S1 Fig in S1 File). By contrast to the changes observed in mitochondrial-dependent oxygen consumption, there was no change in non-mitochondrial oxygen consumption (as the traces converged at the end of the OCR profile) or in the glycolytic flux, upon Tet3-silencing (S3 Fig in S1 File).

We further determined the protein levels of some of the enzymatic components of the OxPhos complexes by immune-blot analyses using a total OxPhos rodent antibody cocktail. Perhaps surprisingly, we found that (by contrast to the transcriptome data) these protein analyses showed no reduction in levels upon the downregulation of TET3 expression (Fig 2E and 2F and see Fig 1C and 1D). Rather, the protein levels of each of the components assessed (ATP5A, UQCRC2, mtCO1, SDHB and NDFUB8) were found to be slightly *increased* upon Tet3-silencing, although the corresponding mRNAs were all slightly decreased. Further, although the individual changes in each component were not found to reach statistical significance, it should be noted that the increases in the proteins of the group when taken as a whole *were* statistically significant in a paired t-test (p = 0.02). The explanation for these unexpected data is not clear but might suggest the involvement of some compensatory pathways, or potentially indicate that the half-lives of the mitochondrial OxPhos proteins are increased in the Tet3-silenced cells (see discussion).

## Tet3-silencing resulted in global hypo-hydroxymethylation

We next investigated the effect of TET3 depletion upon global levels and distribution of 5hmC in N2A cells. HydroxyMeDIP- (hMeDIP)-Seq analyses were performed on biological duplicate samples of genomic DNA from control or Tet3-silenced N2A cells (Arraystar, Inc). The numbers of mapped reads that associated with annotated genetic loci were used for peak detection

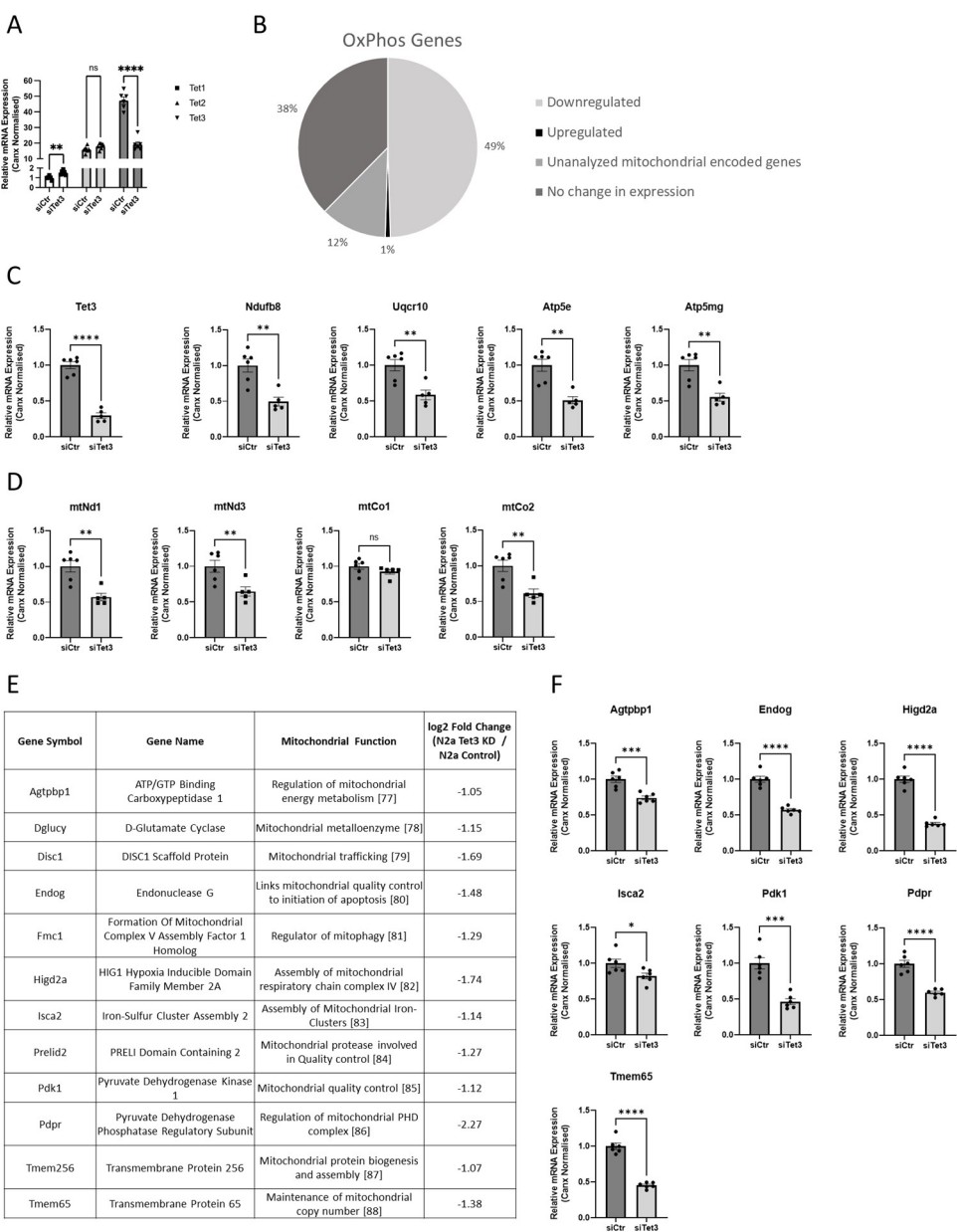

**Fig 1. Silencing of Tet3 acts to decrease mRNA expression of OxPhos and other mitochondrial-related genes in N2A cells.** (A) Relative mRNA expression of Tet1, Tet2 and Tet3 in N2a cells in which Tet3 was silenced for 72 hours and in negative control siRNA-transfected N2a cells (n = 3). (B) Pie chart showing the percentage of the OxPhos genes which were downregulated, upregulated, unchanged or were not analysed in RNA-seq analyses of these cells. (C) Relative mRNA expression of Tet3 and of nuclear OxPhos or (D) mitochondrial-encoded OxPhos genes as indicated, in Tet3-silenced or control N2A cells (n = 6). (E) Table showing mitochondrial function-associated genes that exhibited > 2-fold downregulation in RNA-seq analyses of N2A cells after Tet3-silencing. (F) Relative mRNA expression of selected mitochondrial function associated genes in Tet3-silenced and control N2a cells (n = 6). mRNA expression levels were determined by Q-PCR and normalised to the expression of Canx in all cases. Quantified data are expressed as mean ± SEM, relative to the negative control siRNA samples. Analysed by an unpaired t-test. ns = not significant, * $p < 0.05$, ** $p < 0.01$, *** $p < 0.001$, ****$p < 0.0001$ [77–88].

of hMeDIP-enriched regions across the genome. In both the control and Tet3-silenced groups, a similar percentage of mapped reads were found across promoter regions, gene bodies and

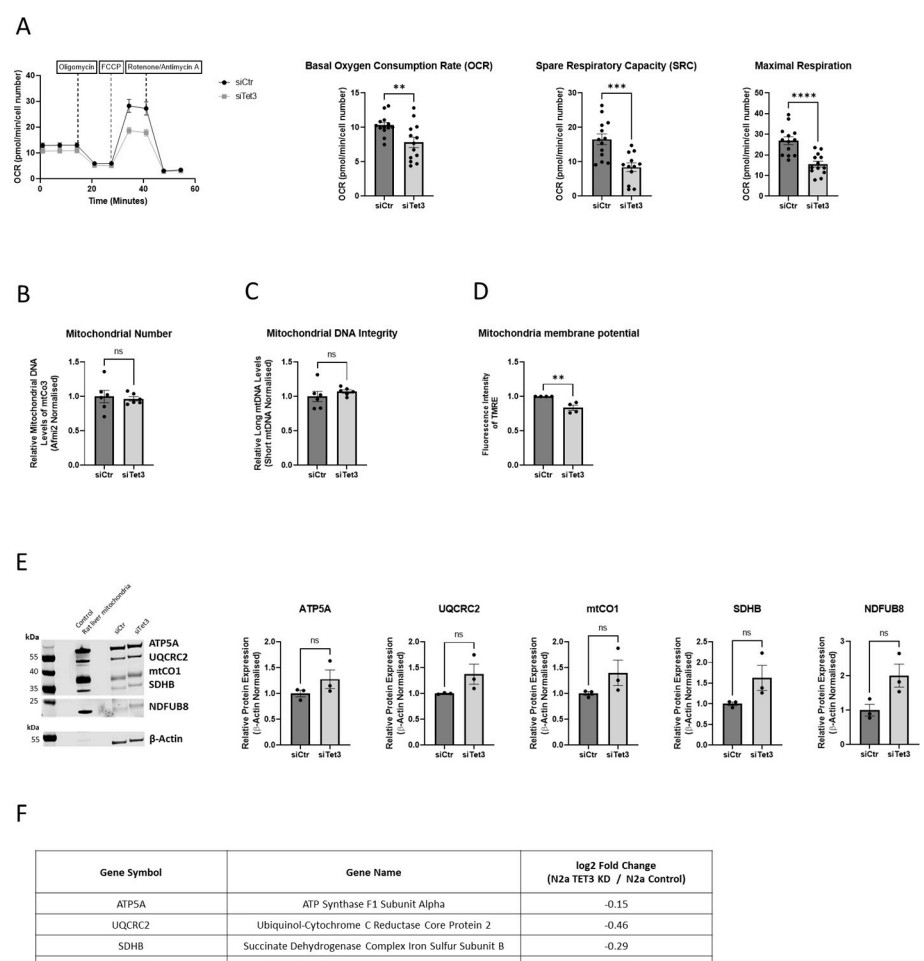

**Fig 2. Silencing of TET3 acted to decrease mitochondrial respiration in N2A cells.** (A) Seahorse SF Mitochondrial Stress Test profile of oxygen consumption rate in N2a cells in which Tet3 was silenced for 72 hours compared to controls. Tet3-silenced cells exhibited reduced basal oxygen consumption rate, spare respiratory capacity and maximal respiration (n = 13 including 3 biological replicates). (B) Mitochondrial number and (C) Mitochondrial DNA integrity in Tet3-silenced, compared to control N2A cells were determined by QPCR (n = 6). (D) Mitochondria potentiation in TET3-silenced and control N2a cells was determined by Fluorescence-activated cell sorting of TMRE-stained cells, normalised to the number of live cells (n = 4). (E) Representative Western blot analyses of OxPhos proteins: ATP5A, UQCRC2, mtCO1, SDHB and NDFUB8 in Tet3-silenced and control N2a cells. Data are shown normalised to the expression of β-Actin (n = 3). To observe NDFUB8 a longer exposure time was carried out. Quantified data are expressed as mean ± SEM, relative to negative control siRNA samples. Analysed by an unpaired t-test ns = not significant, ** $p < 0.01$, *** $p < 0.001$, **** $p < 0.0001$. Although there was no significant change in the expression of any of the individual proteins in E, when taken as a group and analysed by a paired t-test, a significant increase in protein expression upon TET3-depletion was evident (p = 0.02). (F) Table showing change in expression of (nuclear-expressed) OxPhos genes upon Tet3 silencing, as determined by RNA-seq analyses.

intergenic regions (see Fig 3A). Thus, no significant change in the overall genetic distribution of 5hmC deposition, resulting from Tet3-silencing, was evident.

Differentially hydroxymethylated regions (DhMRs) were determined and assigned to the promoter regions of mRNAs, or small or long non-coding RNAs together with enhancer- or superenhancer-associated intergenic regions. In all these regions there was a significantly larger number of hypo-hydroxymethylated, compared to hyper-hydroxymethylated sites in the Tet3-silenced genomic samples, consistent with a catalytic role of TET3 in the conversion of 5mC to 5hmC in these cells (Fig 3B). The promoters (as defined by sequence within 2kb of a

transcriptional start site) of 225 mRNA-expressing genes were found to be hypo-hydroxy-methylated upon TET3 depletion (see S4 Table in S1 File), but there was no evident overlap between these genes and the 172 genes that were significantly transcriptionally down-regulated upon Tet3-silencing in our RNA-seq screen, (including the mitochondrial-associated genes described above that were included in this list). Conversely, there was no overlap between significantly upregulated mRNAs and hyper-hydroxymethylated mRNA gene promoters identified in our study (shown in S5 Table in S1 File). However, 5hmC deposition has been shown to be more functionally relevant with respect to the regulation of enhancers, rather than promoters [28]. We therefore carried out Gene Ontology (GO) analyses on the annotated genes known to be associated with the enhancers and superenhancers which displayed differential hydroxymethylation deposition (Fig 3C), using available data from VISTA (enhancer.lbl.gov)/ FANTOM (fantom.gsc.riken.jp). Although associated gene data were available for only a subset of these enhancer regions, it is striking that in the case of the differentially hypohydroxy-methylated enhancers, 7 of the top 10 identified biological processes are involved in cellular metabolism.

As has been found in other studies [29], the mitochondrial genome was found to be marked with a distinct pattern of 5hmC deposition, but no mitochondrial gene-associated DhMRs were identified to be correlated with the loss of TET3, and the visualised peak signal profiles of the mitochondrial chromosome from the control and Tet3-silenced N2A cells appeared essentially equivalent (see Fig 3D).

The lack of an association between Tet3 downregulation and the loss of 5hmC at the OxPhos and mitochondrial-related gene promoters might suggest that TET3 exerts transcriptional regulation on these genes *via* indirect and/or non-catalytic mechanism(s). To investigate this further, N2A cells were cultured in the DNMT inhibitor, 5-azacytidine (5-azaC), to remove 5mC marks passively through successive rounds of replication. The presence of 5mC, at least at gene promoters, is associated with gene repression [30], thus the loss of such 5mC would result in gene activation. As seen in Fig 3E, with the exception of EndoG (see below), all the OxPhos and mitochondrial genes tested showed no increase in transcriptional expression after 5-azaC administration and in one case (Uqcr10), a significant decrease was observed.

## TET3-dependent regulation of EndoG

As stated above, the mRNA levels of EndoG *did* increase significantly after 5aza-C treatment, consistent with DNA methylation playing a role in the regulation of its transcriptional expression. To further determine whether the catalytic activity of TET3 might be involved in this regulation, the effects of ascorbic acid and dimethyloxalylglycine (DMOG) upon EndoG mRNA expression in N2A cells was investigated. As members of the 2-OGDD superfamily, the activities of TETs are dependent upon the availabilities of both $Fe^{2+}$ and 2-oxoglutarate (2-OG). Ascorbic acid is an antioxidant which acts to reduce $Fe^{3+}$ to $Fe^{2+}$ and may also act as a co-factor of TET enzymes, and accordingly has been shown to increase TET activity [31, 32]. Conversely, as a competitive antagonist of 2-OG, DMOG can be used to inhibit the catalytic action of TETs [33]. Ascorbic acid treatment of N2A cells resulted in a (small but) significant increase of EndoG mRNA, while administration of DMOG significantly decreased it (Fig 4B). Further, upon silencing of Tet3, DMOG did not have any further repressive effect. Taken together, these data are consistent with TET3 regulating EndoG *via* its catalytic activity. In further support of EndoG being a direct regulatory target of TET3, the protein levels of ENDOG were also shown to be downregulated significantly after Tet3-silencing (Fig 4A).

DhMRs within the EndoG gene promoter were not identified to reach the statistical cut off of significance in our hMeDIP-seq analyses. Therefore, we further investigated specifically the

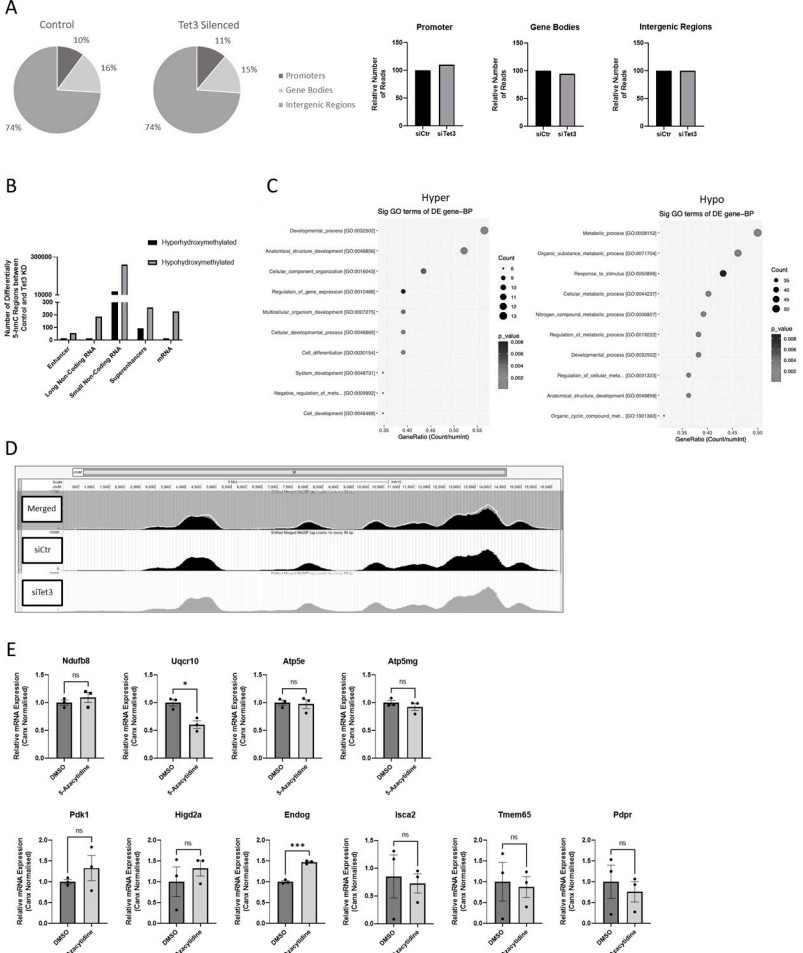

**Fig 3. TET3 has catalytic activity in N2A cells.** Genomic DNAs from control and (72-hour) Tet3-silenced N2A cells were analysed by hMeDIP-seq (Arraystar). Statistically significant enriched regions (peaks) in each case were annotated by the nearest gene using the UCSC RefSeq database, and their positions were classified as promoter, genebody or intergenic. (A) The proportions of the annotated peaks in promoters, gene bodies and intergenic regions were calculated and were essentially equivalent between the control and TET3-depleted cell groups. (B) Differentially hydroxymethylated regions (DhMRs) which demonstrated statistical significance within the promoter regions of mRNA, and long- and short-non-coding RNAs, and within enhancer and superenhancer associated intergenic regions were identified. Tet3-silencing resulted in significantly less hyperhydroxymethylated (5-hmC-enriched regions) and significantly more hypohrydoxymethylated (less enriched) regions, on all these regulatory elements consistent with a loss of TET3-mediated hydroxymethylation (p≤0.01). (C) Functional BP (Biological Process) classification of genes associated with the hyper- and hypo-hydroxymethylated enhancer and superenhancer regions by Gene Ontology (GO) analysis (10 most significant are depicted). Count; the number of differentially-enriched (DE) genes associated with the listed GOID; GeneRatio: The GOID's Gene Ratio Value (equal to Count/Total gene list). (D) 5-hmC enrichment in the mitochondrial genome of TET3-silenced and control N2a cells were visualised using UCSC Genome Browser (https://genome.ucsc.edu/). The pattern of enrichment of 5-hmC varies across the mitochondrial genome but is similar between TET3-silenced and control N2a cells. (E) Relative mRNA expression of nuclear-encoded OxPhos genes; NdufbB8, Uqcr10, Atp5e and Atp5mg and mitochondrial-function-associated genes; Pdk1, Higd2a, Endog, Isca2, Tmem65 and Pdpr in N2a cells treated with 5-azaC (1µM) for 72 hours and in vehicle control (DMSO)-treated cells were determined by Q-PCR, normalised to the relative expression of Canx. (n = 3). Quantified data are expressed as mean ± SEM, relative to vehicle control DMSO samples. Analysed by an unpaired t-test. ns = not significant, * p<0.05, *** p<0.001.

5hmC deposition over the whole EndoG gene locus to determine whether there might be DhMR regions within the gene body. From the visualised peak signal profiles two regions were

identified which displayed differential 5hmC enrichment (see hatched boxes in Fig 4C). One of these was located over the first intron/exon boundary and overlapped with a CpG island. The other was situated over the intron 2/exon 3 boundary. To identify TET3-dependent changes in 5hmC at base resolution, an enzyme-based conversion followed by sequencing was undertaken [25]. T4 Phage β-glucosyltransferase (T4-BGT) specifically transfers the glucose moiety of uridine diphosphoglucose (UDP-Glc) to 5-hmC. The beta-glucosyl-5-hydroxy-methylcytosine thus formed are then specifically protected from the action of APOBEC; a family of cytosine deaminases which convert cytosines to uracils. PCR-based amplification and sequencing of specific regions can then identify specifically 5hmC, as distinct from both 5mC and unmodified cytosine. We investigated the 5hmC status of specific sequences within the regions identified above, by amplification and sequencing of Epi P1 and Epi P2 (Fig 4C). Epi P1 included some sequence identified as a CpG island, but no evidence for 5hmC deposition in this amplicon was evident either in the control or Tet3-silenced cells (S4 Fig in S1 File). By contrast, sequencing of the Epi P2 revealed a considerable number of sites within a specific region (depicted as a striped box) where conversion to uracil was incomplete in the untreated (control) cells, whereas these cytosines were fully converted in the Tet3-silenced samples. This indicates the presence of 5hmC in the untreated N2A cells that were lost upon Tet3-silencing. Moreover, the majority of these modified cytosines are found within CpA, rather than CpG, dinucleotides (Fig 4D).

## Discussion

TET proteins and the initial product of their catalytic function; 5hmC, have increasingly been shown to be functionally important both in neurodevelopment and adaptation, and in the aetiology and progression of age-related neurodegenerative diseases [34–37]. TET3, specifically, has been shown to play crucial roles in cognitive brain function such as memory and learning (potentially in a sex-specific manner [9, 38]) and therefore presents as a potential therapeutic target in the treatment of neurodegenerative disease. However, at the cellular level, the functional consequences and gene targets of TET3-specific transcriptional regulation and the mechanisms underlying this regulation in neurons remain to be elucidated.

We addressed this by analysing the effects of Tet3 genetic silencing upon the transcriptome of the neuroblastoma-derived cell line, N2A. Of the three mammalian TET proteins, TET1 is most highly expressed and functionally relevant in embryonic stem cells [39]. By contrast, TET2 and TET3 are typically co-expressed in many somatic cells and gene ablation studies in mice have suggested that they may have overlapping functions and are in some situations able to compensate for each other's loss [40]. In N2A cells we observed no (compensatory) increase in Tet2 mRNA levels, upon Tet3-silencing and only a small increase in (the relatively low level of endogenous) Tet1 expression (Fig 1A), while global 5hmC levels were reduced. Further, an increasing number of studies have shown the TET proteins to perform both functionally and mechanistically distinct roles [41, 42]. We therefore conclude that the significant changes that we observed in the epigenome, transcriptome and cellular function are TET3-dependent.

We identify here a crucial role of TET3 in the regulation of efficient mitochondrial respiration in N2A cells. The downregulation of TET3 resulted in a robust decrease in mitochondrial-dependent respiratory activity. Thus, basal respiration, maximal respiration and spare respiratory capacity were all reduced in the TET3-depleted cells, while non-mitochondrial respiration was unaffected. N2A cells are tumour-derived and a potential criticism of the findings of our study is that their metabolism might be more dependent on glycolysis than OxPhos [43]. However, in neuronal cells it is well recognised that a substantial metabolic shift from anaerobic glycolysis to fatty acid and glucose oxidation is associated with differentiation

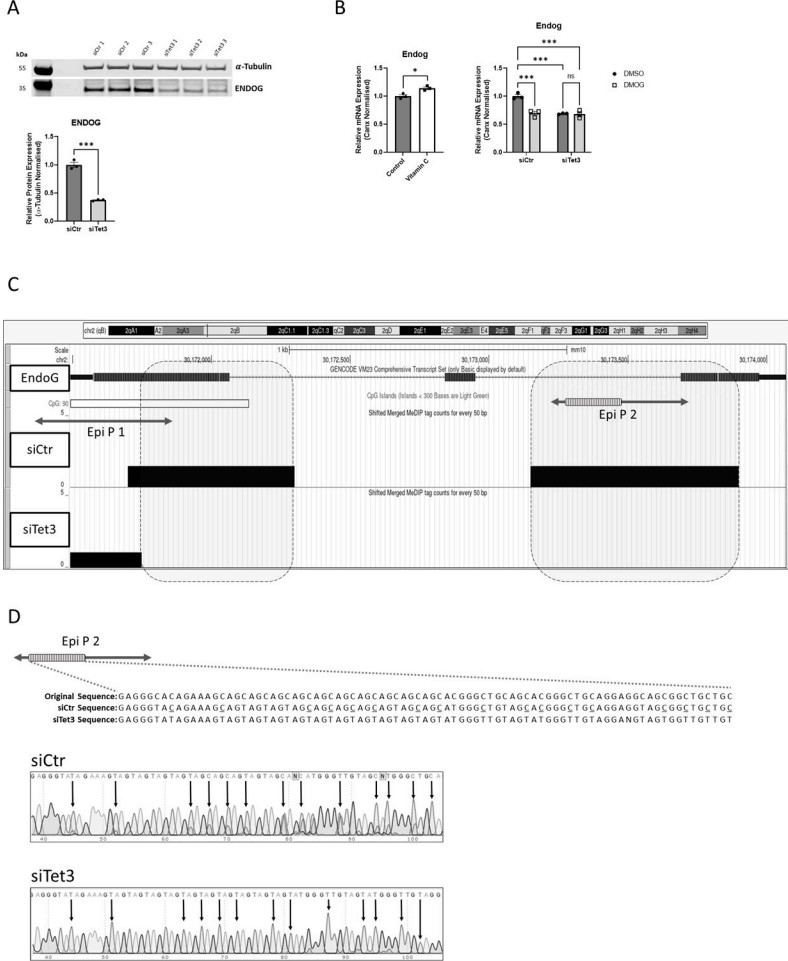

**Fig 4. EndoG is a catalytic target of TET3.** (A) Relative protein expression of TET3 in N2a cells in which Tet3 was silenced for 72 hours and in negative control siRNA-transfected N2a cells (n = 3). (B) Relative mRNA expression of EndoG in N2a cells treated with vitamin C (50 mM; an activator of 2OGDDs) for 24 hours, compared to control cells and in Tet3-silenced or control N2a cells, both treated with either DMOG (1mM; an inhibitor of 2OGDD enzymes) for 48 hours or vehicle control (DMSO). Levels were determined by Q-PCR, normalised to the relative expression of CANX. (n = 3). Quantified data are expressed as mean ± SEM, relative to vehicle control DMSO and control samples. Analysed by an unpaired t-test and two-way ANOVA with Tukey's multiple comparison test. ns = not significant, * p<0.05, *** p<0.001. (C) 5-hmC enrichment within the EndoG gene locus was visualised using UCSC Genome Browser (https://genome.ucsc.edu/) on hMeDIP-seq data of Tet3-silenced and control N2a cells. The signals (black lines) represent the extended read counts in every 50bp region. The grey hatched boxes highlight the loss of 5-hmC enrichment (black) between siCtr (middle section) and siTet3 (bottom section) within the EndoG (grey, top section) gene locus in N2a cells. A CpG island is represented with a white box. Enrichment of 5-hmC is lost at the 5' end over the boundary of the coding region of the first exon and the first intron, including some of the CpG island region and at the 3' end over the last intron/exon boundary. Locations of the amplicons that were generated and sequenced (EpiP1 and Epi P2) are shown as two-sided arrows. (D) The sequence of a region of Epi P2 after amplification from Tet3-silenced or control N2a cells. 5hmC residues within control, N2A genomic DNAs were protected from deamination by APOBEC, by conversion to glucosylated-hydroxymethylcytosine by treatment with T4 Phage β-glucosyltransferase. The original sequence is depicted. Unprotected cytosines are read as thymine and cytosines partially protected from conversion in the (SiControl) N2A genome are underlined. After Tet3-silencing all cytosines are read as thymine. The absence of any cytosines (indicated by black arrows) in Tet3-silenced N2a cells suggests the presence of 5-hmC in the control N2A cells is lost after Tet3 silencing.

during development [44]. Further, the impairment of mitochondrial function, specifically in N2A cells, has been shown to impair their differentiation capacity [45] suggesting that OxPhos is functionally relevant in these cells and therefore validates their usage here.

The reduction in mitochondrial respiration correlated with reduced mRNA levels of both nuclear- and mitochondrial-expressed OxPhos components. However, we found no evidence for TET3-mediated, catalytic demethylation-dependent transcriptional regulation of the OxPhos components either in our hMeDIP screen, or as determined by a (lack of) activation by 5-azaC administration. Further, the lower mRNA levels of OxPhos components did not correlate with lower protein levels. Rather, the levels of the OxPhos proteins tested were all found to increase slightly, and although individually none of these changes were found to be significant (n = 3, p>0.05 in all cases), when taken as a group, the combined increases were statistically significant. The phenomenon of mitochondrial proteotoxicity; the reduction in bioenergetics due to the accumulation of dysfunctional proteins, can arise from an increase in expression and/or reduction in degradation of aberrant mitochondrial proteins [46]. Multiple mitochondrial quality control pathways (such as mitophagy, mitochondrial biogenesis and mitochondrial dynamics) [47] act to maintain mitochondrial homeostasis. The finding that several proteins known to be involved in these mechanisms were significantly downregulated at the transcriptional level by TET3 depletion (see Fig 1) might therefore support a role for TET3 in such homeostasis. Also consistent with TET3 acting to maintain mitochondrial quality (rather than number), is the observation that the mitochondrial membrane potential ($\Delta\Psi$m) was found to be significantly reduced in the Tet3-silenced N2A cells. Maintenance of $\Delta\Psi$m is essential for key cellular functions such as cell proliferation, production of reactive oxygen species (ROS) and oxygen sensing [48] and the loss of $\Delta\Psi$m triggers pathways to clear damaged mitochondrial components. Conversely, a lack of mitochondrial quality control mechanisms leads to impaired turnover (and consequent increased half-life) of mitochondrial proteins and aberrant protein accumulation in the inner mitochondrial membrane leading to progressive dissipation of $\Delta\Psi$m [49]. It has also been shown in yeast that a loss of $\Delta\Psi$m leads to a transcriptional downregulation of OxPhos components [50]. We would therefore suggest that the transcriptional downregulation of both nuclear- and mitochondrial-expressed OxPhos components that we observed in the TET3-depleted N2A cells might be *via* an indirect mechanism, as a downstream consequence of impaired mitochondrial quality control.

Other studies have investigated the effects of Tet3-ablation on the transcriptome of *mature*, hippocampal neurons and identified differential expression of genes related to synapses and synaptic transmission [51]. Further, striking differences in the transcriptional function of TET3 in dorsal, compared to ventral, hippocampal neurons have been reported, suggesting that the molecular function(s) of TET3 in (predominantly terminally-differentiated) neurons are highly specific to their neuronal roles [9]. Metabolic processes are known to underlie the differentiation and function of neuronal cells [52]. We subjected the (publicly available) differential transcriptome data set of dorsal adult hippocampal cells from Tet3 KO and control mice (GSE 140850) [9] to Ingenuity Pathway Analysis. Perhaps significantly, after "Synaptogenesis Signalling", "Oxidative Phosphorylation" and "Mitochondrial Dysfunction" were 2 of the next 4 most highly regulated pathways (P = 3.45E-03 and 3.9E-03 respectively). We therefore suggest that the role of TET3 in the regulation of mitochondrial respiration identified here may likely also impact upon the functions of TET3 in mature neurons.

In common with other studies, our data are consistent with TET3 exerting direct transcriptional regulatory effects both by catalytic and non-catalytic mechanisms. A loss of TET3 clearly resulted in overall hypo-hydroxymethylation at all the (annotated) regulatory regions investigated, consistent with other reports of the catalytic enzymatic activity of TET3 in some specific neuronal cells [53] including N2As [54]. However, most of the transcriptional changes that we observed were not readily found to correlate with DhMRs, and there was no overlap found between significantly transcriptionally-changed genes (p≤0.0001) and genes identified with DhMRs within their promoters. However, (hydroxy)methylation changes have been implicated in mediating longer-range *cis* regulatory effects and chromatin interactions which act upon specific gene transcription [28, 55]. Intriguingly, GO analysis of genes which associated with enhancer regions exhibiting significantly less 5hmC deposition upon Tet3-silencing, identified many (associated) metabolic processes as likely biological functions. This might suggest TET3 to play a wider role in the regulation of cellular metabolism in neuronal cells.

By contrast to the OxPhos genes, EndoG (a gene also observed to be downregulated upon TET3 depletion (P = 0.015) in (adult) mouse dorsal hippocampi (accession no.GSE 140850) [9]), was identified as a direct catalytic target of TET3 in N2A cells. In addition to the transcriptional (and protein) expression of EndoG being downregulated upon TET3-depletion, it was also shown to be upregulated by 5azaC or ascorbic acid treatment and down-regulated by DMOG treatment (in a TET3-dependent fashion). Crucially, we demonstrated specific 5hmC marks within the EndoG gene body that were lost upon silencing of Tet3. In addition to 5hmC deposition at enhancers, the significance of the accumulation of these marks within gene bodies (and in particular its positive correlation with gene transcription) is now being investigated. Very recent evidence suggests that the intragenic generation of 5hmC by TET3, specifically, may act to stabilize full-length gene transcripts by preventing aberrant Pol II-dependent initiations within gene bodies [56]. It is also noteworthy that most of the TET3-dependent 5hmC marks, observed within the EndoG locus were at CpA, rather than at CpG, dinucleotides. Levels of non-CpG methylation are known to be exceptionally high in mammalian brains, compared to other tissue and organs, and to play a critical role(s) in the regulation of cognitive function, *via* the binding of the transcriptional regulator; MeCP2 [57]. TET3 has been shown to be capable of binding to both non-CpG- and CpG-containing DNA oligonucleotides *in vitro* [58], while TET3 depletion was demonstrated to result in hypermethylation, preferably at non-CpG sites in a chicken B-cell line [59]. Our data might further suggest a role for TET3 in the regulation of non-CpG methylation and hydroxymethylation in neuronal cells, potentially linked to cognitive function. A future investigation of the potential functional relationship between TET3, non-CpG sites, and the regulation of transcriptional initiations within gene bodies would therefore be of clear interest.

Functionally, ENDOG is a nuclease, which is mainly located in the mitochondrial intermembrane space and can be released from the mitochondria and translocated to the nucleus where it induces genomic DNA fragmentation leading to apoptosis [23]. Several studies have identified the functional significance of ENDOG in both immature, neuroblastoma cells and in mature neurons in the regulation of cell death after ischeamic/hypoxic injury [60–63] and in excitotoxity [64, 65]. Typically, rapid nuclear translocation of ENDOG occurs in response to insult/injury and the forced reduction of ENDOG in such pathophysiological settings has proved beneficial. In some normal physiological contexts, such as in neural development and neuroplasticity, the involvement of (mitochondrial-dependent) apoptosis is evident (reviewed in [66]), but the potential role for ENDOG in mediating cell death in these settings has not, to our knowledge, thus far been investigated.

More recently, a role for nuclear-translocated ENDOG in the activation of the DNA damage response (DDR) to enhance autophagy has been demonstrated [67]. Perhaps consistent

with our findings, a functional role for TET3, specifically, in the regulation of the DDR in N2A cells, that was dependent upon its catalytic activity, has been shown previously [54]. Further, the requirement of autophagy for ENDOG-dependent DNA-damage repair is regulated by TET activity [68]. This raises the intriguing possibility that the enzymatic activity of TET enzyme(s) may be involved in both the regulation of expression and function of ENDOG in DNA-damage repair. In this regard it may be significant that ENDOG preferentially recognises and cleaves 5hmC-modified DNA [69].

An impaired DDR (and consequent damage accumulation) is linked causally to mitochondrial dysfunction [70] and roles for both mitochondrial and nuclear aberrant DNA repair mechanisms have been linked to mitochondrial dysfunction in neurodegenerative diseases such as Parkinson's (reviewed in [71]. Further, the role of critical role of ENDOG in physiological mitochondrial bioenergetics *in vivo* has been demonstrated, as deletion of Endog in mice resulted in hearts with depleted and dysfunctional mitochondria, resulting in impaired mitochondrial respiration [72]. In this regard it has long been suggested that ENDOG may play a critical role in mitochondrial DNA replication and repair (which would clearly impact upon OxPhos), and the recent observation of mutations in EndoG that associate with mitochondrial myopathy and multiple mtDNA deletions would strongly support this [73].

It should, however, be noted that we observed changes in many other genes, known to be involved in mitochondrial function, upon TET3 downregulation (Fig 1E), albeit these changes were likely regulated by a different (non-catalytic) mechanism. We therefore suggest that the clear downregulation of ENDOG, apparent upon Tet3-silencing (Fig 4A), may contribute in part to the mitochondrial phenotype, but is not likely, in itself, to be sufficient to cause the changes in OxPhos observed here.

## Concluding remarks

Mitochondrial dysfunction is considered to be the key and common factor for the pathologies of (age-related) neurodegenerative diseases [74, 75] and the TET proteins, including TET3, together with the epigenetic mark that they generate (5hmC) play significant roles in the development of these diseases [34]. We show here that TET3 is an important regulator of mitochondrial function and that EndoG is a direct target of its catalytic activity. Given the known roles of ENDOG in the DDR [67, 68] and the emerging link between DDR linked to mitochondrial dysfunction and neurodegenerative diseases [71, 76], these findings may therefore prove to be of future clinical importance.

## Supporting information

**S1 File. Contains all details of primers, FACS gating strategy, 5hmC dot blots, glycolytic capacity of N2A cells, converted sequencing data, lists of differentially expressed genes and differentially hydroxymethylated regions and uncropped Western blots.**
(PDF)

## Acknowledgments

We thank Dr John Pizzey for critical reading of this manuscript.

## Author Contributions

**Conceptualization:** Valeria Leon Kropf.

**Data curation:** Youwen Yang.

**Investigation:** Valeria Leon Kropf, Caraugh J. Albany, Hannah L. H. Green.

**Methodology:** Anna Zoccarato.

**Supervision:** Caraugh J. Albany, Anna Zoccarato, Alison C. Brewer.

**Writing – original draft:** Alison C. Brewer.

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
