## [Decision Letter · Decision Letter 0]

21 Sep 2023

PONE-D-23-26086TET3 is a positive regulator of mitochondrial respiration in neuronal cellsPLOS ONE

Dear Dr. Brewer,

Thank you for submitting your manuscript to PLOS ONE. After careful consideration, we feel that it has merit but does not fully meet PLOS ONE’s publication criteria as it currently stands. Therefore, we invite you to submit a revised version of the manuscript that addresses the points raised during the review process. Mostly positive, the reviewers have raised a few concerns, especially on the presentation and interpretation of some data.  A reviewer has concerns regarding the title of the manuscript. The authors are advised to address those concerns in their revised manuscript. 

We look forward to receiving your revised manuscript.

Kind regards,

Jyotshna Kanungo, Ph.D.

Academic Editor

PLOS ONE

Journal Requirements:

   "This work was supported by the National Program of Scholarships and Student Loans (PRONABEC), Minister of Education, Peru (Law No 29837, Supreme Decree 013-2012-ED, Supreme Decree 008-2013-ED, Supreme Decree 01–2015-MINEDU, RJ No 4320-2018-MINEDU-VMGI-PRONABEC. We thank Dr John Pizzey for critical reading of this manuscript"

   "This work was supported by the National Program of Scholarships and Student Loans (PRONABEC), Minister of Education, Peru (Law No 29837, Supreme Decree 013-2012-ED, Supreme Decree 008-2013-ED, Supreme Decree 01–2015-MINEDU, RJ No 4320-2018-MINEDU-VMGI-PRONABEC to VLK.  The sponsors played no role in the study design, data collection and analysis, decision to publish, or preparation of the manuscript."

Additional Editor Comments:

Mostly positive, the reviewers have raised a few concerns, especially on the presentation and interpretation of some data. The authors are advised to address those concerns in their revised manuscript.

Reviewers' comments:

Reviewer's Responses to Questions

**Comments to the Author**

1. Is the manuscript technically sound, and do the data support the conclusions?

Reviewer #1: Partly

Reviewer #2: Yes

2. Has the statistical analysis been performed appropriately and rigorously? 

Reviewer #1: Yes

Reviewer #2: Yes

3. Have the authors made all data underlying the findings in their manuscript fully available?

Reviewer #1: Yes

Reviewer #2: Yes

4. Is the manuscript presented in an intelligible fashion and written in standard English?

Reviewer #1: Yes

Reviewer #2: Yes

5. Review Comments to the Author

Reviewer #1: This study used pharmacological and genetic manipulations, as well as transcriptomic and biochemical analysis to investigate the role and targets of TET3 in N2A cell line. The result from RNA seq suggested that oxidative phosphorylation was identified as the most significantly down-regulated pathway by Tet3-silencing, consisted with the downregulation of oxygen consumption rate, the mRNA levels of nuclear- and mitochondrial encoded oxidative phosphorylation, and mitochondrial quality control genes. Among these genes, EndoG was identified as a direct target of TET3-catalytic activity at non-CpG methylated sites within its gene body. The authors indicated that the decrease in OxPhos may be due to dysfunctional mitochondria rather than direct transcriptional regulation of OxPhos genes by TET3. This work represents enriched data and a series of interesting observations. However, there are some major concerns regarding the interpretation of the results.

Major

1. The protein levels of each of the components assessed (ATP5A, UQCRC2, mtCO1, SDHB and NDFUB8) were found to be slightly increased upon Tet3-silencing, in contrast to the transcriptome data and the Q-PCR verification. Does Tet3-silencing enhance the protein synthesis or decrease the protein degradation? It is easy to assess by different inhibitors.

2. Figure 4 Restoring the full-length Tet3 or the kinase-dead mutants should be included to elucidate the specific role of TET3 on the EndoG regulation in N2a cells.

3. What is the functional relationship of TET3 and EndoG in regulating the mitochondrial respiration or oxidative phosphorylation in N2A cell? Synergistically or not? Does restore EndoG rescue the impaired phenotype by TET3 silencing?

4. The title is misleading because N2A is neuroblastoma cell line.

5 “The decrease in OxPhos may be due to dysfunctional mitochondria rather than direct transcriptional regulation of OxPhos genes by TET3” is overstated.

Reviewer #2: This is a well written manuscript with some minor issues:

1. Please define "low glucose" in the methods, with mM concentration for the N2a cells.

2. Please define the concentrations of drugs used in the seahorse assay (oligomycin, FCCP, and Rotenone/Antimycin A)

3. In the results section page 6, 2nd paragraph there is a highlighted XXX which needs to be filled in with the correct information for the GEO accession number.

6. PLOS authors have the option to publish the peer review history of their article (what does this mean?). If published, this will include your full peer review and any attached files.

Reviewer #1: No

Reviewer #2: **Yes: **Heather Wilkins

---

## [Author Response · Author response to Decision Letter 0]

20 Oct 2023

We would like to thank the reviewers for taking the time to review our manuscript and for their helpful suggestions.

We have now amended our manuscript as detailed below. 

Reviewer #1: This study used pharmacological and genetic manipulations, as well as transcriptomic and biochemical analysis to investigate the role and targets of TET3 in N2A cell line. The result from RNA seq suggested that oxidative phosphorylation was identified as the most significantly down-regulated pathway by Tet3-silencing, consisted with the downregulation of oxygen consumption rate, the mRNA levels of nuclear- and mitochondrial encoded oxidative phosphorylation, and mitochondrial quality control genes. Among these genes, EndoG was identified as a direct target of TET3-catalytic activity at non-CpG methylated sites within its gene body. The authors indicated that the decrease in OxPhos may be due to dysfunctional mitochondria rather than direct transcriptional regulation of OxPhos genes by TET3. This work represents enriched data and a series of interesting observations. However, there are some major concerns regarding the interpretation of the results.

Major

1. The protein levels of each of the components assessed (ATP5A, UQCRC2, mtCO1, SDHB and NDFUB8) were found to be slightly increased upon Tet3-silencing, in contrast to the transcriptome data and the Q-PCR verification. Does Tet3-silencing enhance the protein synthesis or decrease the protein degradation? It is easy to assess by different inhibitors.

Response: We fully concur that both enhanced (aberrant) protein synthesis and reduced protein degradation could result in the discrepancy between the changes in mRNA and protein levels of the OxPhos that we observed upon TET3 downregulation together with the decrease in mitochondrial bioenergetics. This point is now discussed (highlighted in the 4th para of discussion). However, we have not, as yet, carried out any further experiments (using inhibitors) to investigate this further. We believe that due to the complex nature of the different pathways which regulate mitochondrial function this would prove quite challenging and time consuming. We believe that this kind of mechanistic study (based on our findings here) might be more informative (and potentially clinically relevant) if performed in primary neurons, which is beyond the scope of this paper. 

2. Figure 4 Restoring the full-length Tet3 or the kinase-dead mutants should be included to elucidate the specific role of TET3 on the EndoG regulation in N2a cells.

Response: We identify EndoG as a catalytic target of TET3 as we show it to be upregulated by 5-azaC (suggesting it is regulated by methylation), to be downregulated by DMOG (indicating it to be regulated by the catalytic activity of a 2-OGDD enzyme), and to exhibit changes in 5-hmC upon TET3 downregulation. The rescue experiments proposed, using wild-type and kinase (catalytic?)-dead mutants would be inappropriate in our system, since we deplete TET3 expression using siRNAs, which would also degrade the “rescuing” gene expression. In addition, TET3 in mouse has both long and short isoforms, and it is not clear which would be the relevant isoform to test functionally in the rescue experiment. However, this is, of course a very interesting question with respect to the biochemistry of TET3, and determining which isoform is responsible for the catalytic activity on the EndoG promoter in a neuronal-like cell is something that we are currently aiming to determine. 

3. What is the functional relationship of TET3 and EndoG in regulating the mitochondrial respiration or oxidative phosphorylation in N2A cell? Synergistically or not? Does restore EndoG rescue the impaired phenotype by TET3 silencing?

Response: The silencing of TET3 resulted in the significant downregulation of a number of genes, known to be involved in mitochondrial function (Fig 1E &F). One purpose of our study was to determine which genes might be regulated by the catalytic activity of TET3, and only EndoG fell into that category. However, the other genes regulated by other (non-catalytic/more downstream) mechanisms would also contribute to the functional dysregulation of the mitochondria. Thus, we do not suggest that EndoG misexpression alone is responsible for the loss of mitochondrial bioenergetics, or that Endo G expression alone would rescue the phenotype. Rather, we suggest that it may be involved. However, we now realise that this was not made sufficiently clear and we apologise. We have now included a paragraph at the end of the discussion to clarify this point. 

4. The title is misleading because N2A is neuroblastoma cell line.

Response: We have now changed the title of the manuscript to make it clear that the study was performed in Neuro2A cells.

5 “The decrease in OxPhos may be due to dysfunctional mitochondria rather than direct transcriptional regulation of OxPhos genes by TET3” is overstated.

Response: We have changed this sentence (at the end of the abstract) to “Accordingly, we propose that aberrant mitochondrial homeostasis may contribute to the decrease in OxPhos, observed upon Tet3-downregulation in Neuro2A cells”. We believe that this makes it clearer that the changes in OxPhos observed may in part also result from the changes in OxPhos gene expression.

Reviewer #2: This is a well written manuscript with some minor issues:

1. Please define "low glucose" in the methods, with mM concentration for the N2a cells.

Response: This information is now included in the methods section.

2. Please define the concentrations of drugs used in the seahorse assay (oligomycin, FCCP, and Rotenone/Antimycin A)

Response: This information is now included in the methods section.

3. In the results section page 6, 2nd paragraph there is a highlighted XXX which needs to be filled in with the correct information for the GEO accession number.

Response: If our manuscript is accepted, we will provide the repository information. We have also informed the editor of this.

---

## [Decision Letter · Decision Letter 1]

27 Oct 2023

TET3 is a positive regulator of mitochondrial respiration in Neuro2A cells

PONE-D-23-26086R1

Dear Dr. Brewer,

We’re pleased to inform you that your manuscript has been judged scientifically suitable for publication and will be formally accepted for publication once it meets all outstanding technical requirements.

Kind regards,

Jyotshna Kanungo, Ph.D.

Academic Editor

PLOS ONE

Additional Editor Comments (optional):

Reviewers' comments:

Reviewer's Responses to Questions

**Comments to the Author**

1. If the authors have adequately addressed your comments raised in a previous round of review and you feel that this manuscript is now acceptable for publication, you may indicate that here to bypass the “Comments to the Author” section, enter your conflict of interest statement in the “Confidential to Editor” section, and submit your "Accept" recommendation.

Reviewer #1: All comments have been addressed

2. Is the manuscript technically sound, and do the data support the conclusions?

Reviewer #1: Yes

3. Has the statistical analysis been performed appropriately and rigorously? 

Reviewer #1: Yes

4. Have the authors made all data underlying the findings in their manuscript fully available?

Reviewer #1: Yes

5. Is the manuscript presented in an intelligible fashion and written in standard English?

Reviewer #1: Yes

6. Review Comments to the Author

Reviewer #1: My questions have been addressed appropriately by the authors in the revision. I have no additional comments.

7. PLOS authors have the option to publish the peer review history of their article (what does this mean?). If published, this will include your full peer review and any attached files.

Reviewer #1: No

---

## [Editor Report · Acceptance letter]

21 Nov 2023

PONE-D-23-26086R1 

TET3 is a positive regulator of mitochondrial respiration in Neuro2A cells 

Dear Dr. Brewer:

I'm pleased to inform you that your manuscript has been deemed suitable for publication in PLOS ONE. Congratulations! Your manuscript is now with our production department. 

Kind regards, 

on behalf of

Dr. Jyotshna Kanungo 

Academic Editor

PLOS ONE